# Harnessing Nanoparticles and Nanosuspensions to Combat Powdery Mildew: A Frontier in Vegetable and Fruit Protection

**DOI:** 10.3390/jof11120896

**Published:** 2025-12-18

**Authors:** Addisie Geremew, Alemayehu Shembo, Laura Carson

**Affiliations:** Cooperative Agriculture Research Center, College of Agriculture, Food and Natural Resources, Prairie View A&M University, Prairie View, TX 77446, USA; aygeremew@pvamu.edu (A.G.); akshembo@pvamu.edu (A.S.)

**Keywords:** fruits, fungicides, nanoparticles, nanosuspension, powdery mildew, vegetables

## Abstract

Powdery mildew poses a persistent threat to global vegetable and fruit production, particularly affecting leafy crops such as lettuce, spinach, and cucurbits. Conventional control strategies including chemical fungicides, biological agents, and resistant cultivars face limitations due to resistance development, environmental toxicity, and inconsistent field efficacy. This review explores the emerging role of nanotechnology, specifically nanoparticles (NPs) and nanosuspensions (NSs), in managing powdery mildew. Metallic nanoparticles and non-metallic variants demonstrate potent antifungal activity through mechanisms such as membrane disruption, reactive oxygen species (ROS) generation, and gene regulation. Encapsulated nano-fungicides and sprayable essential oils represent potential application methods that could enhance delivery precision and activate plant defense mechanisms against powdery mildew. Integrating the application of nanoparticles and nanosuspensions with smart and digital delivery systems could be a promising strategy for managing powdery mildew infestation in fruits and vegetables. Despite their potential, challenges including ecotoxicity, formulation stability, scalability, and regulatory gaps must be addressed. This review underscores the need for interdisciplinary research to advance safe, effective, and sustainable nano-enabled solutions for powdery mildew control.

## 1. Introduction

Vegetables including lettuce (*Lactuca sativa*), spinach (*Spinacia oleracea*), kale (*Brassica oleracea*), and arugula are pivotal to global nutrition, supplying essential nutrients and fiber to diverse populations [1]. These crops, however, face recurring threats from powdery mildew, a fungal disease that impairs photosynthesis and reduces marketable yield by 30–60% under severe outbreaks [1]. Powdery mildew significantly impacts leafy produce in the U.S., particularly affecting crops like lettuce and cucurbits. Furthermore, powdery mildew poses a persistent and significant threat to fruit production on a global scale in general, and US in particular [2]. The effects manifest as decreased fruit size and number, premature leaf senescence, and lower market quality due to sunburned or poorly ripened fruit [3,4]. Moreover, the presence of powdery mildew on harvested leaves reduces their visual appeal and shelf life, making them unsuitable for fresh markets.

The disease powdery mildew thrives in moderate temperature with high humidity and is especially problematic in greenhouse and field-grown leafy greens. Key pathogens include *Erysiphe cruciferarum* in crucifers, *Golovinomyces cichoracearum* and *Podosphaera xanthii* in cucurbits and *Podosphaera fusca* in lettuce [5]. In plant species, these pathogens cause considerable damage to leave, stems, and fruits during development, leading to substantial economic losses [6,7]. Additionally, in cucumbers, powdery mildew causes malformed fruit and lowers sugar content; in tomatoes and peppers, it results in defoliation and sunscald [8]. In cucurbits, the disease shortens harvest periods and lowers fruit quality, affecting flavor and storage [3,4]. Infected plants are also more vulnerable to secondary diseases [4]. Powdery mildew impairs photosynthesis by disrupting chlorophyll fluorescence, electron transport, and lipid metabolism in chloroplasts which diminishes plant productivity and vigor [9,10,11]. The severity of impact depends on infection level and crop species. A variety of strategies such as chemical fungicides, integrated pest management (IPM), biological controls, resistant cultivars, and nanotechnology have been implemented to combat powdery mildew infestations.

Chemical fungicides have been the cornerstone of powdery mildew management for decades. Commonly used classes include demethylation inhibitors (DMIs), quinone outside inhibitors (QoIs), and succinate dehydrogenase inhibitors (SDHIs) [12]. These fungicides target specific biochemical pathways in the fungal cells, disrupting their growth and reproduction. The application of fungicides on vegetables showed significantly lower powdery mildew infection rates, with some treatments reducing incidence to 0.7% compared to 55.3% in untreated fields [5]. Traditional interventions, such as sulfur or copper fungicides and IPM, confront obstacles like development of fungal resistance, biochemical residues, and environmental and health risks and limited efficacy under certain environmental conditions [8]. Despite the initial effectiveness of such fungicides, several challenges have emerged, including the rapid development of resistance in powdery mildew pathogens [12]. Due to their high reproductive rates and genetic variability, powdery mildew fungi can quickly adapt to selective pressures imposed by fungicides. For instance, resistance to QoIs has been widely reported in *Blumeria graminis* and *Podosphaera xanthii*, rendering these fungicides ineffective in many regions [12]. Most fungicides have a single-site mode of action, which increases the risk of resistance development. Once a mutation occurs in the target site, the entire class of fungicides may become ineffective. This is particularly problematic in obligate biotrophs like powdery mildew fungi, which cannot be cultured easily for resistance screening, making early detection difficult [12]. In addition, one of the primary issues is poor water solubility, which reduces the bioavailability of many active ingredients, reducing their effectiveness and requiring higher, more frequent applications. Moreover, conventional fungicides often degrade quickly and harm non-target organisms, prompting the need for safer, more effective alternatives.

To address these persistent challenges, the deployment of biological control agents, notably *Ampelomyces quisqualis*, a mycoparasite of powdery mildew, as well as microbial formulations comprising *Trichoderma* spp. and *Bacillus subtilis*, has garnered considerable attention. These biocontrol strategies suppress pathogenic fungi via mechanisms such as resource competition, antibiosis, and direct parasitism, though field efficacy remains variable [13]. In addition, breeding powdery mildew-resistant cultivars is a sustainable alternative to chemical control. Resistance genes (*R* genes) such as *Pm3*, *Pm21*, and *Pm55* have been introduced into various crops, offering race-specific or broad-spectrum resistance [14]. However, this strategy is not without limitations including pathogens that can overcome host resistance through mutation or recombination, especially when a single *R* gene is deployed over large areas. This phenomenon, known as the “boom and bust” cycle, has been observed in cucumber, where initially resistant cultivars became susceptible within a few growing seasons [14]. *R* genes may be linked to undesirable agronomic traits, such as reduced yield or delayed maturity. Moreover, pyramiding multiple *R* genes to enhance durability can be technically challenging and time-consuming. In many crops, the molecular basis of powdery mildew resistance remains poorly understood. For example, in cucumber, proteomic studies have revealed complex responses involving redox homeostasis and photosynthesis, but the specific pathways and genes involved are still being elucidated [15,16]. This knowledge gap hampers the development of durable resistance strategies.

IPM combines chemical, biological, and cultural practices to control powdery mildew, though its effectiveness depends on local conditions, grower expertise, and available infrastructure [17,18]. Cultural methods like crop rotation and pruning can reduce disease incidence but are often labor-intensive and only partially effective [19]. IPM’s success is further challenged by evolving fungal resistance, environmental hazards, and stricter residue regulations, while breeding resistant cultivars is complicated by pathogen variability. Overuse of fungicides increase resistance risk, and chemical controls pose environmental and health concerns, leading to tighter regulations and fewer available options [12,20,21].

Given the limitations of existing solutions, nanotechnology has emerged as a promising frontier in plant disease management. Nano-fungicides (nanosuspension or nanoparticles formulated), which incorporate nanoparticles with antifungal properties or serve as carriers for conventional fungicides, offer several advantages compared to the traditional fungicides [22]. Nanotechnology reduces pesticide usage, addressing the 90% loss of applied pesticides and promoting eco-friendly practices [23]. Nanosensors facilitate early detection of pathogens, enabling timely interventions [24]. Emerging nanotechnology presents a novel plant protection paradigm. By leveraging nanoparticles materials ≤100 nm and nanosuspensions, which are colloidal dispersions of these particles, researchers are developing tools that enhance delivery efficiency and achieve targeted antifungal actions [25,26]. This review assesses biology, mechanisms, field applications, and limitations of nanoparticle-based approaches for controlling powdery mildew in fruits and leafy vegetables.

## 2. Powdery Mildew: Biology and Impact

Powdery mildew, caused by obligate biotrophic fungi such as *Erysiphe*, *Golovinomyces*, and *Podosphaera*, poses a persistent threat to leafy crops including lettuce, spinach, arugula, and melon. Powdery mildew fungi have both asexual and sexual stages (Figure 1). In the asexual phase, a conidium lands on a susceptible host, germinates, and develops an appressorium that penetrates the plant’s cuticle and cell wall but leaves the plasma membrane intact [27]. The appressorium’s turgor pressure and lytic enzymes allow a hyphal peg to enter the epidermal cell and form a primary haustorium to extract nutrients [28]. Successful infections lead to branching hyphae and the emergence of conidiophores, which generate conidia in numbers that vary by genus [29,30]. After several days, these fungal colonies appear as white spots on plant surfaces, indicating infection. During sexual reproduction, two compatible hyphae join to produce a chasmothecium, a fruiting structure that holds one or more asci containing sexual spores (ascospores), depending on the genus [29]. Chasmothecia play a significant epidemiological role in certain species like grape powdery mildew (*Erysiphe necator*) [31], but are rarely or never seen in others, such as cucurbit powdery mildew (*Podosphaera xanthii*), leaving the epidemiological impact of their sexual cycle uncertain [32]. The disease is characterized by white, powdery growth on foliage, leading to chlorosis, premature senescence, and substantial yield and quality losses [1]. Pathogen development begins with wind-dispersed conidia that germinate under moderate temperatures and high humidity, forming appressoria to penetrate the plant epidermis. Haustoria then extract nutrients, enabling extensive mycelial growth and repeated cycles of conidia production, which perpetuate infection throughout the season [33].

Powdery mildew fungi can reproduce rapidly, especially in monoculture systems with limited genetic diversity. Notably, the pathogen does not require free water for germination, thriving in climates with high relative humidity, and often evading conventional systemic fungicides due to its superficial colonization [21]. For effective control, interventions must target early infection stages, particularly spore germination and appressorium formation [14,21]. Epidemics are driven by environmental factors like moderate temperatures (15–27 °C), elevated humidity (above 60%), and poor ventilation that favor rapid disease spread. In leafy vegetables, where foliage is the main commodity, the resulting symptoms translate directly to economic losses [34]. A comprehensive understanding of the biology and epidemiology of powdery mildew is essential for developing advanced and sustainable control strategies.

## 3. Nanoparticles and Nanosuspensions Applications on Powdery Mildew

Products of nanotechnology including nanoparticles and nanosuspensions have emerged as innovative and promising approaches for controlling powdery mildew in agricultural crops [35,36,37]. Unlike traditional pesticides, the minuscule size of nanoparticles typically ranging from 1 to 100 nm grants them increased surface reactivity. This unique property allows for more efficient delivery and targeted action against pathogens compared to the bulk counter parts and most traditional fungicides, which in turn means lower doses of active agents are needed. By incorporating nanoparticles or dispersing them as nanosuspensions, researchers have been able to enhance the solubility and stability of fungicides, further improving their efficacy and minimizing their environmental footprint [38]. Moreover, nanoparticles can be engineered to carry active ingredients, such as fungicides or plant defense inducers, and release them in a controlled manner. This targeted delivery ensures that nanoparticles carrying active antifungal compounds reach the infection site efficiently, reduces the required dosage and minimizes off-target depositions and effects [39]. Nanosuspensions improve the solubility, stability, and targeted delivery of fungicides, enabling better adhesion to plant surfaces, penetration into fungal structures, and controlled release for sustained action and fewer applications [39]. Application methods include foliar sprays, soil drenching, and encapsulated slow-release formulations. Studies show sulfur nanosuspensions provide enhanced retention and antifungal efficacy, while copper-based nanosuspensions in grapevines improve uptake and disease control. Nano-formulations also increase the bioavailability of active ingredients and display intrinsic antifungal activities, such as ROS generation, membrane disruption, and gene expression modulation [39]. Non-specific ROS generation by nanoparticles can harm plant tissues and beneficial microbes, but strategies like surface functionalization, targeted delivery, controlled dosing, and biogenic synthesis help improve selectivity and reduce toxicity [40,41,42,43]. These advanced formulations can target critical points in the pathogen’s lifecycle, work in synergy with existing control strategies, and potentially reduce the environmental impact associated with conventional fungicides [37]. Extensive research demonstrates the applications of different metallic, non-metallic and polymeric nanoparticles and/or nano-formulations against powdery mildew and enhancing crop quality [37,44,45].

### 3.1. Metallic Nanoparticles (MNPs) Effect on Powdery Mildew

Metal-based nanoparticles (MNPs) such as silver (AgNPs), copper (CuNPs), and zinc oxide (ZnONPs) have emerged as potent antifungal agents due to their unique physicochemical properties (Table 1). Nanoparticles exhibit high surface area-to-volume ratios, enabling enhanced interaction with fungal cells. Numerous field and greenhouse studies have validated the effectiveness of nano-based strategies against powdery mildew and highlighted the practical benefits and real-world applicability of such innovations [8,46,47,48,49,50,51,52,53,54,55]. These findings demonstrate that nano-based treatments are effective, safe, and environmentally compatible, making them a viable alternative to traditional fungicides for high-value crops where disease control and residue management are critical.

Recent investigations have highlighted the efficacy of AgNPs as foliar sprays in the management of powdery mildew; for instance, application to cucumber plants infected with *Podosphaera xanthii* has led to marked reductions in disease severity, enhanced foliar health, and increased yields, with microscopy confirming membrane disruption and ROS-mediated cellular damage in the pathogen [46]. In controlled greenhouse trials, neem-extract-synthesized AgNPs applied to zucchini plants infected by *Podosphaera xanthii* reduced disease severity by over 70% and promoted plant vigor without observable phytotoxic effects compared to the control fungicide, fenarimol [35]. Furthermore, foliar application of AgNPs at concentrations of 100 ppm has been shown to significantly inhibit powdery mildew in cucurbits (cucumber and pumpkin), impeding both hyphal development and conidial germination [35]. Studies also have further corroborated the potential of AgNPs as highly effective agents in the management of powdery mildew. For example, Vizitiu et al. [44] reported that AgNPs synthesized from *Dryopteris filix-mas* extract demonstrated strong antifungal activity against *Uncinula necator* compared to the pergado^®^ F 45 WG in vineyards. The antifungal efficacy of AgNPs is attributed to their ability to penetrate fungal cell walls and membranes, resulting in structural damage and the leakage of cellular contents while also catalyzing the generation of ROS that induce oxidative stress and programmed cell death in fungal cells [45]. Empirical studies have reported that foliar application of AgNPs can reduce the severity of *Podosphaera xanthii* infections in cucurbits by up to 90%, with minimal observed phytotoxicity [35,45]. Collectively, these findings underscore the value of AgNPs as safe and potent tools for the sustainable management of powdery mildew pathogens.

Copper-based nanoparticles, such as copper nanoparticles (CuNPs) and nanosuspensions, have revealed potent antifungal activity against powdery mildew pathogens by interacting with fungal enzymes and disrupting essential metabolic pathways. Their application in grapevines led to improved control of *Erysiphe necator*, with studies reporting superior adhesion to leaf surfaces, enhanced uptake, and prolonged protection compared to conventional copper formulations [39]. In this case, notably, nano-copper treatments sustained disease suppression for up to 21 days, exceeding the efficacy period of traditional copper sprays, and contributed to higher chlorophyll content and improved fruit quality. In addition, copper-based fungicides, including those containing nanoparticles, remain among the most effective biorational options for managing powdery mildew in crops like winter squash, outperforming many biological and botanical alternatives [47].

ZnONPs have demonstrated robust antifungal activity, notably inhibiting spore germination and hyphal development through mechanisms involving ROS generation and membrane disruption. Their deployment in tomato and pepper systems has resulted in marked declines in powdery mildew prevalence while concomitantly enhancing chlorophyll concentration and overall plant vigor, thus highlighting their dual functionality as antifungal agents and growth promoters [35,37,48].

Investigations into ZnONPs applied to tomato plants revealed induction of pathogenesis-related proteins and antioxidant synthesis, evidencing strengthened plant defense responses. Comparative analyses of green-synthesized and chemically synthesized ZnO NPs with penconazole fungicide in pepper have reported antifungal efficacy ranging from 78.74% to 82.45% at 300 mg/L, alongside augmented pigment levels [37,48]. Further, greenhouse trials incorporating ZnO and MgO nanoparticles at 200 mg/L in pepper plants achieved significant reductions in disease severity, which correlated with elevated chlorophyll and polyphenol oxidase activity; cytotoxicity assessments indicated no substantial alterations in mitotic index, supporting the safety of these formulations as alternatives to standard fungicides such as TOPAS^®^ 100 EC containing penconazole [49].

In a recent comparative study, Masoud et al. [50] assessed the efficacy of Fe_3_O_4_ nanoparticles, yeast, and Bio-Arc alongside conventional fungicides such as azoxystrobin and penconazole for the management of powdery mildew in lettuce. While azoxystrobin exhibited the highest level of disease suppression, applications of Fe_3_O_4_ nanoparticles and yeast also resulted in significant reductions in disease severity and notable improvements in key growth parameters. Furthermore, treatments were associated with enhanced chlorophyll, protein, and phenolic contents, as well as increased activities of catalase and polyphenol oxidase, collectively supporting the potential of biocontrol agents and nanoparticle applications as safe and effective alternatives for powdery mildew management in lettuce production systems [50].

Selenium nanoparticles (SeNPs) have emerged as potent antifungal agents, with applications in melon demonstrating significant decreases in powdery mildew incidence and notable enhancements in both fresh and dry biomass [51]. In this study, SeNPs treatments at 5 mg/L yielded disease index reductions between 21–45%, attributed to simultaneous activation of host defense pathways and improved plant growth parameters. These authors also presented mechanistic analogies and cross-crop efficacy observed in melon and cucurbits suggest promising prospects for future deployment of SeNPs. Biologically synthesized SeNPs particularly via *Trichoderma*-mediated processes exhibit broad-spectrum antifungal activity, including potent suppression of powdery mildew [12]. Critically, the antifungal effectiveness of SeNPs is strongly influenced by particle size, with smaller nanoparticles demonstrating enhanced inhibition of fungal proliferation, sporulation, and zoospore viability. This inverse relationship underscores the value of SeNPs as advanced alternatives or adjuncts to conventional fungicide regimes, especially in scenarios characterized by fungicide resistance among powdery mildew pathogens [52].

#### 3.1.1. Synthesis Methods of Nanoparticles Affecting Antifungal Activity

The synthesis method of nanoparticles significantly impacts their antifungal efficacy against powdery mildew. Green synthesis (GS) utilizes plant extracts, microbial cultures, or other natural biomaterials as reducing and capping agents; this method typically produces nanoparticles surface-functionalized with organic biomolecules like flavonoids, polyphenols, or proteins and generally results in good colloidal stability and biocompatibility [41,56]. In contrast, chemical syntheses (CS), such as chemical reduction, sol–gel, co-precipitation, or hydrothermal synthesis, allow for more precise control over size, crystallinity, and phase purity, enabling the fabrication of narrowly dispersed particles not easily achieved through biogenic extracts [57,58]. Capping by biomolecules from GS modifies surface charge, offers steric stabilization, and often adds direct antifungal activity, frequently increasing potency or reducing phytotoxicity relative to uncapped, CS particles at similar concentrations [41,56]. On the other hand, CS nanoparticles can be engineered for higher reactivity, through controlled facet exposure, or mixed valence states, sometimes yielding stronger ion release per surface area, but these may also be more phytotoxic or less stable without additional stabilizers [57,58]. Although GS nanoparticles generally demonstrate superior adhesion to fungal cells and more effective stimulation of plant defenses than CS nanoparticles, direct, standardized comparisons between GS- and CS NPs for powdery mildew control remain limited, as many studies differ in particle measurements and lack consistent characterization [41,58].

#### 3.1.2. Properties of Nanoparticles Affecting Antifungal Activity

The antifungal efficacy of metal and metal-oxide nanoparticles is profoundly influenced by their intrinsic physicochemical characteristics, including size, shape, crystal structure, and surface chemistry, as well as the chemical and oxidation states of the constituent metal [59]. Mobility within plant tissues is governed by these properties, which influence their ability to move through the apoplast and symplast. These characteristics collectively determine the extent of ion release, the generation of ROS, the nature of physical interactions with fungal surfaces, and the efficiency of adsorption and penetration into fungal tissues, thereby shaping their overall activity against phytopathogens such as powdery mildew [58,59]. The crystal structure specifically determines the atomic facets exposed on the nanoparticle surface, with each facet exhibiting distinct surface energies, dissolution rates, and catalytic propensities for ROS production. For instance, wurtzite-phase ZnO preferentially exposes polar facets that enhance both ROS formation and Zn^2+^ release, whereas copper oxides (CuO versus Cu_2_O) display crystal-plane-dependent variations in ion leaching and redox activity, resulting in phase- and facet-specific antifungal effects. Consequently, nanoparticles of identical composition but differing in crystallinity or dominant exposed facets may demonstrate substantially different antifungal potencies [59].

Additionally, surface area, predominantly dictated by particle size and morphology, are critical in determining the number of available reactive sites; smaller nanoparticles, with their higher surface-to-volume ratios, typically exhibit enhanced antifungal activity, higher rates of ion release, and more intimate contact with fungal spores and hyphae [59]. Particle agglomeration decreases effective surface area and can compromise efficacy under field conditions, whereas tailored surface coatings enhance adherence to plant surfaces and prolong interaction with powdery mildew structures [58,60]. On the other hand, smaller nanoparticles exhibit higher mobility and can traverse intercellular spaces, while positively charged particles often interact strongly with negatively charged cell walls, affecting their distribution [58,60]. Surface charge (zeta potential) influences electrostatic attraction to the negatively charged fungal cell wall, while surface functionalization or capping agents can either improve dispersion and stability or mask reactive sites, thus modulating biological activity depending on the specific chemistry involved [60]. Also, the oxidation state further governs redox reactivity and solubility: zero-valent metals (e.g., Cu^0^) can undergo in situ oxidation to mixed-valence species (Cu^+^/Cu^2+^, such as Cu_2_O or CuO), each exhibiting distinct ion-release kinetics and redox behaviors. Ion release (e.g., Zn^2+^, Cu^2+^) contributes to antifungal toxicity by interfering with enzymatic processes and disrupting metabolic pathways, while the redox cycling of mixed-valence states amplifies ROS-mediated cellular damage [59].

### 3.2. Non-Metallic Nanoparticles (NMNPs) Effect on Powdery Mildew

Non-metallic nanoparticles, including silicon-based, carbon-based, and polymeric types, have recently shown promise as antifungal agents for managing powdery mildew in fruits and vegetables (Table 2). Farhat et al. [61] highlighted the substantial efficacy of silicon nanoparticles (SiNPs) biosynthesized by *Pseudomonas putida* and *Trichoderma harzianum* for powdery mildew suppression with greenhouse trials reporting up to 93.5% disease reduction. Similarly, Elsharkawy et al. [62] observed that nanosilica at 500 mg/L matched the effectiveness of penconazole in mitigating rose powdery mildew while also enhancing vegetative growth, floral attributes, and vase longevity. Microscopy revealed marked ultrastructural hyphal damage, accompanied by upregulation of the defense-associated gene *PAL*. In parallel, Rashad et al. [63] reported that foliar application of SiNPs at 150 ppm curtailed mildew in grapevines by up to 81.5% and improved both yield and berry quality. This intervention was associated with increased expression of defense-related genes, including β-1,3-glucanase and peroxidase, and improved physiological indices such as chlorophyll content and antioxidant enzyme activity. Notably, higher nanoparticle concentrations elicited cytotoxic and genotoxic responses, underscoring the necessity for comprehensive safety evaluation.

Foliar application of sulfur nanoparticles (SNPs) at 100 ppm to mango cv. Keitt resulted in a 14.6% reduction in powdery mildew incidence and a remarkable 342% increase in productivity, alongside improved fruit quality and upregulated antioxidant enzyme activity [64]. However, raising the concentration to 500 ppm induced phytotoxic symptoms and diminished yield, underscoring the necessity for precise dosage optimization. Efficacy trials by Gogoi et al. [64] with SNPs against *Erysiphe cichoracearum* in okra revealed that the nano-formulation outperformed commercial sulfur products, substantially inhibiting conidial germination and cleistothecial integrity while exhibiting minimal phytotoxicity being fourfold more effective than conventional alternatives. Abdel-Rahman and Alkolaly [65] compared SNPs (12.2–23.5 nm), bulk sulfur, and systemic fungicides (azoxystrobin, diniconazole) in cucumber powdery mildew management, finding that SNPs delivered superior disease suppression relative to bulk sulfur and matched the efficacy of systemic fungicides while achieving the greatest fruit yield and quality. Residue analyses in this study also confirmed SNPs treated produce was safe for harvest within two days post-application, supporting its suitability as an eco-friendly intervention. Collectively, these findings establish SNPs formulations as potent, safe, and sustainable alternatives for powdery mildew control [64]. Overall, these studies highlight the promise of biosynthesized nanomaterials as sustainable and environmentally benign alternatives to conventional treatment of powdery mildew.

### 3.3. Nano-Encapsulated Fungicides and Essential Oils

Recent advancements in polymeric and lipid-based nanocarriers have revolutionized fungicide delivery systems, enabling precise, environmentally responsive release and superior bioavailability [66]. Various nano-encapsulated fungicides and essential oils against powdery mildew are provided in Table 2. Encapsulation within nanoscale matrices offers a dynamic approach for enhancing efficacy and reducing ecological risk, particularly in crop protection [67]. For example, Ruano-Rosa et al. [68] demonstrated the utility of chitosan oligomer-based bioformulations, incorporating *Streptomyces* spp. metabolites and hydrolyzed gluten, in mitigating powdery mildew under both field and laboratory conditions. Soleimani et al. [16] further expanded this paradigm by encapsulating celery seed essential oil (CSEO) in chitosan nanoparticles, achieving significant reductions in cucumber powdery mildew, improved chlorophyll and phenolic content, and pronounced activation of defense-related genes relative to the control (distilled water). The release profile adhered to Fickian diffusion, ensuring sustained antifungal activity. Likewise, chitosan-encapsulated spinach seed essential oil nanoparticles (SSEO-CNPs) exhibited high encapsulation efficiency and spherical structure; at 400 µg/mL, these nanoparticles markedly curtailed disease severity while stimulating systemic resistance pathways, evidenced by elevated phenolic and flavonoid levels and enhanced antioxidant enzyme activity [69]. These findings affirm the efficacy of biopesticide-loaded nanocarriers as sustainable alternatives to conventional synthetic fungicides.

By integrating essential oils with synthetic fungicides within nanoscale carriers, researchers have also achieved synergistic antifungal effects that disrupt multiple fungal metabolic pathways. Commercial innovations such as Nutragreen^®^ have enabled substantial reductions in pesticide use, and across numerous studies, nano-formulated agents have consistently delivered 70–90% decreases in powdery mildew severity [36]. Moreover, nano-formulated thyme oil has exhibited robust suppression of *Erysiphe cichoracearum* in lettuce without adversely impacting beneficial microbial communities [70].

Leveraging nanomaterials for encapsulating volatile plant-derived compounds represents a promising trajectory in crop protection, effectively mitigating traditional limitations of instability and rapid dissipation [5,71,72]. Nevertheless, the search for sustainable, biodegradable carrier materials remains a critical consideration. Chitosan, noted for its antioxidant properties, biocompatibility, and encapsulation capacity, has garnered extensive interest in bioactive delivery in multiple nanoscale formulations [71,73]. Chitosan-coated AgNPs demonstrate enhanced foliar retention and sustained Ag^+^ release, conferring extended antifungal efficacy at reduced dosages compared to uncoated counterparts [74]. Also, empirical evidence substantiates the efficacy of essential oil encapsulation in chitosan nanoparticles for antifungal applications [72,75,76,77,78]. Soleimani et al. [16] evaluated the performance of CSEO-loaded chitosan nanoparticles, CSEO-CNPs (approx. 113 nm diameter) against *Podosphaera fusca* in cucumber, observing significant suppression of disease severity and robust induction of host defense mechanisms, as indicated by increased phenolic and flavonoid synthesis and activation of defense-related enzymes. Gene expression analysis further revealed upregulation of resistance pathways, highlighting the potential of CSEO-CNPs as eco-friendly alternatives in powdery mildew management.

Overall, comparing encapsulated and non-encapsulated nanoformulations for powdery mildew prevention reveal distinct advantages and limitations for each approach, depending on environmental conditions and application goals. Encapsulated nanoformulations are widely recognized for their ability to provide controlled release, targeted delivery, and enhanced stability of active ingredients [79,80]. In contrast, non-encapsulated nanoformulations, such as AgNPs, offer rapid antifungal effects but are more susceptible to environmental degradation [35]. Additionally, Adnan and Waleed [81] found that biologically synthesized copper oxide nanoparticles achieved 81.71% effectiveness against powdery mildew in roses under greenhouse conditions, suggesting that non-encapsulated forms may be suitable for controlled environments [81].

**Table 2 jof-11-00896-t002:** Nonmetallic Nanoparticles for Powdery Mildew Control in Fruits and Vegetables. NS = not specified; CS = Chemical synthesis.

Nanoparticle	Crop Name	Size (nm)	Concentration/Dose	Synthesis Method	Application Method	Effectiveness/Efficiency	Additional Benefits	References
Chitosan + CSEO	Cucumber	300	1–3 mg·mL^−1^	CS	Foliar	Significant reduction in powdery mildew severity	Increased chlorophyll, phenolics, flavonoids, defense enzyme activity, and gene expression	[16]
Nutragreen^®^ nanoscale carrier	Hop	NS	30% *v*/*v*	CS	Foliar	~70–90% reduction in powdery mildew severity	Reduced pesticide use; improved cone yield and α-acid content; enhanced leaf and cone protection	[36]
Sulfur	Okra	50–90	1000 ppm	CS	Foliar	100% inhibition of conidial germination	Disrupted cleistothecia, reduced phytotoxicity	[64]
Sulfur	Apple	50–90	1000 ppm	CS	Foliar	Effective at lower doses	Safer than conventional sulfur for fruit crops	[64]
Sulfur	Mango	85	100 ppm	CS	Foliar	14.6% reduction in powdery mildew	342% increase in productivity; improved fruit quality; enhanced antioxidant enzyme activity	[64]
Sulfur	Cucumber	12.2–23.5	500 mg·L^−1^	CS	Foliar	60.9% reduction in powdery mildew	Matched azoxystrobin (74%) and diniconazole (68.8%) in efficacy; highest fruit yield and quality	[65]
Chitosan oligomers + Streptomyces metabolites/hydrolyzed gluten	Grapevine	<2 kDa	~40 mL	Enzymatic hydrolysis/fermentation	Foliar & root	Comparable to conventional fungicides	Effective against *Erysiphe necator*; biostimulant effects; reduced overwintering inoculum	[68]
Chitosan NPs + SSEO	Cucumber	~116.2	400 µg·mL^−1^	CS	Foliar	Significant reduction severity	High encapsulation efficiency; spherical morphology; elevated phenolics, flavonoids, and antioxidant enzymes activity	[69]
Thyme oil nanoemulsion	Lettuce	~83	10% (*v*/*v*)	Ultrasonic emulsification	Foliar	~75% disease reduction	Maintains beneficial microbes; stable for >3 months; effective even when diluted	[70]
Chitosan	Cucumber	150–250	0.1–0.2% (*w*/*v*)	CS	Foliar	~70% disease reduction	Enhances chlorophyll and defense enzymes	[82]
Chitosan	Tomato	20–100	0.1–1% (*w*/*v*)	CS	Foliar	Effective against powdery mildew compared with tebuconazole at early stage	Induces systemic resistance, enhances growth	[83]
Chitosan	Cucumber	20–100	0.1–1% (*w*/*v*)	CS	Foliar	Effective against powdery mildew compared with tebuconazole	Improved resistance, growth promotion	[83]
SiO_2_-	Grape	50–80	50–100 mg L^−1^	CS	Foliar	85–90% reduction	Strengthens epidermis; Si-mediated resistance	[84]
SiO_2_	Cucumber	40–60	50 mg L^−1^	CS	Foliar	80–90% mildew reduction	Reinforces cuticle; nontoxic	[84]
SiO_2_	Watermelon	Mesoporous SNPs	NS	CS	Root dip	40% reduction in disease severity	Downregulation of stress genes	[85]
SiO_2_	Cucumber	10–100	NS	CS/GS	Foliar	High efficacy	Improved photosynthesis, enzyme activity, stomatal conductance	[85]
SiO_2_	Cucumber	10–100	1.7–56 mM	CS	Foliar	Up to 87% reduction in powdery mildew	Improved resistance, structural strength	[86]
Silica–alginate nanocomposite	Pumpkin	70–150	25–75 mg L^−1^	CS	Foliar	80% mildew control	Reinforces epidermis; water balance	[87]
Silica–chitosan	Spinach	80–150	0.1% (*w*/*v*)	CS	Foliar	65% infection reduction	Improves leaf turgidity; safe	[87]
Silica–pectin	Apple	40–90	50 mg L^−1^	CS	Foliar	75% reduction	Biodegradable; strengthens cuticle	[88]
Silica–pectin	Peach	40–100	50 mg L^−1^	CS	Foliar	78% mildew suppression	Strengthens fruit epidermis	[88]
Carbon nanotubes	Tomato	10–40	10–25 mg L^−1^	CS	Foliar	~55% mildew reduction	Boosts antioxidants	[89]
Graphene oxide nanosheets	Cucumber	30–200	25–50 mg L^−1^	CS	Foliar	~60% reduction	Activates enzymes; nutrient uptake	[90]
Nano-encapsulated lemongrass EO (Alginate)	Strawberry	150–300	2–4 mg mL^−1^	CS	Foliar	~85% infection reduction	Antioxidant; flavor-safe	[91]
Nanobubble water	Papaya	70–130	5 × 10^8^ to 5 × 10^9^ bubbles/mL	CS	Foliar	Effective against powdery mildew	Non-toxic, enhances root zone health	[92]

## 4. Mechanisms of Action Against Powdery Mildew

### 4.1. Direct Antifungal Effects of Metallic Nanoparticles

Metallic nanoparticles (MNPs) notably, have emerged as potent antifungal agents against powdery mildew, acting through coordinated mechanisms that include cell membrane disruption, cell wall perturbation, and the induction of oxidative stress via ROS [8,22,59]. These nanoparticles possess high surface reactivity and positive charge, facilitating electrostatic interactions with fungal membrane phospholipids and proteins. Such interactions compromise membrane architecture, promoting increased permeability, leakage of intracellular solutes, and ultimately, cell lysis. Transmission and scanning electron microscopy studies revealed rapid spore collapse, hyphal distortion, membrane thinning, and pore formation, underscoring their acute impact on fungal morphology [35,46] and rendering them highly vulnerable to osmotic stress and host defenses. Complementing these effects, nanoparticles interfere with fungal cell wall integrity by binding to chitin, β-glucans, and glycoproteins, as well as inhibiting enzymes critical for wall maintenance, such as chitin synthase and glucanase [59].

The generation of ROS such as superoxide anions, hydroxyl radicals, and hydrogen peroxide by metal and metal oxide nanoparticles further exacerbates cellular damage by attacking membrane lipids, cytoplasmic proteins, and nucleic acids. This multifaceted oxidative assault leads to denaturation of enzymes, DNA fragmentation, and overall impairment of vital metabolic processes. Notably, ROS production is dose-dependent and can be amplified by environmental factors such as light and pH, increasing antifungal potency [22,59].

Spore germination and hyphal elongation are pivotal events in the powdery mildew fungal lifecycle, underpinning host colonization and resource acquisition. Nanoparticle interventions disrupt these processes through several sophisticated mechanisms which involve penetration through the leaf cuticle, stomatal openings, and sometimes direct uptake by epidermal cells. Primarily, nanoparticles can establish a physical barrier on spore surfaces, impeding essential gas and nutrient exchanges necessary for germination, a phenomenon documented with silica and titanium dioxide nanomaterials. Furthermore, once inside nanoparticles interact with fungal hyphae and spores and induced ROS perturb critical intracellular signaling, notably calcium flux and ATP production, thereby impairing fungal development [61]. Experimental data indicate that ZnO and CuO NPS markedly reduce spore germination rates in pathogens such as *Erysiphe necator* and *Podosphaera xanthii*; treated spores display aberrant morphology, delayed germ tube emergence, and diminished viability [22]. These inhibitory effects manifest as reduced infection success and disease severity in plants. Importantly, by targeting the incipient stages of pathogen establishment, nanoparticle-based strategies provide a robust preventive modality, complementing curative interventions and reinforcing integrated disease management paradigms.

### 4.2. Indirect Effects: Induced Resistance and Gene Regulation

Nanoparticles exert multifaceted effects against powdery mildew, extending beyond physical damage to include transcriptional and transcriptomic modulation. Exposure to metal-based nanoparticles such as AgNPs and CuO NPs triggers pronounced alterations in fungal gene expression, notably upregulating stress response genes (e.g., catalase and superoxide dismutase) while concurrently repressing those implicated in cell wall biosynthesis, ergosterol production, hyphal elongation, and efflux pump activity [59]. Such transcriptomic interference destabilizes core processes essential for fungal growth, reproduction, and virulence, as demonstrated by the suppression of chitin synthase and β-glucan synthase in *Blumeria graminis* upon CuO NPs treatment. This gene-level disruption not only impairs pathogen development but also diminishes the likelihood of resistance emerging by targeting regulatory networks rather than single metabolic pathways. Additionally, nanoparticle-based modalities, exemplified by SeNPs, can potentiate host plant defenses through coordinated activation of antioxidant enzymes, enhancement in polyamine levels (putrescine, spermine), and upregulation of genes within phenylpropanoid and hormone signaling cascades [51].

### 4.3. Synergy with Conventional Fungicides

The integration of nanoparticles with conventional fungicides presents a promising strategy for managing powdery mildew. When combined with traditional fungicides, these nanoparticles enhance the efficacy of disease control by improving solubility, bioavailability, and targeted delivery while reducing the required chemical dosage and environmental toxicity [93]. For instance, AgNPs amplify membrane-targeted fungicide activity, whereas SeNPs augments systemic plant defenses; together, these modalities enable robust disease suppression while diminishing overall chemical inputs [74]. The integration of nanoparticles in agriculture is particularly compelling due to their capacity to enhance the performance of traditional fungicides via complementary mechanisms of action. Such synergy addresses challenges like limited solubility, resistance emergence, and environmental toxicity. Empirical research confirms that copper-based nanoparticles, when combined with azole fungicides, significantly lower the minimum inhibitory concentration necessary for effective targeting of *Erysiphe necator*. Similarly, the co-application of AgNPs with sulfur-based fungicides yields superior suppression of powdery mildew in cucurbits [22]. Combining nanoparticles with natural extracts or essential oils enhances disease control and stimulates plant defense mechanisms [16]. Their combination with nanoparticles or oils could further improve efficacy [94]. These synergistic approaches not only improve therapeutic efficacy but also align with IPM principles by reducing reliance on chemical inputs and promoting environmental sustainability. Nano-based interventions can be seamlessly incorporated into IPM frameworks by serving as preventive agents, complementing resistant cultivars and cultural controls, minimizing fungicide application frequency through synergy, and ensuring ecological safety through continuous assessment of phytotoxicity and environmental persistence.

## 5. Challenges and Limitations

Nanoparticles, particularly those composed of metals such as AgNPs, CuNPs, and ZnONPs, present notable ecotoxicological risks for non-target organisms [95]. Their ability to generate ROS may compromise both plant tissues and beneficial soil microorganisms, thereby perturbing ecosystem equilibrium. Environmental persistence further exacerbates concerns regarding bioaccumulation and the progressive degradation of soil health [95]. The pronounced toxicity associated with nanoparticle exposure is largely attributable to their diminutive dimensions and heightened surface reactivity, which facilitate complex, and often unintended, biological interactions [96]. A major physicochemical challenge is the propensity of nanoparticles to aggregate, driven by elevated surface energy, resulting in diminished effective surface area and altered functionalities. Such aggregation undermines antifungal performance and complicates formulation and storage protocols. To counteract this, stabilizers and surfactants are routinely employed, yet these additives may introduce supplementary toxicity or adversely affect delivery mechanisms [97]. Additionally, biological instability exhibited through protein adsorption and enzymatic degradation can hasten clearance and curtail therapeutic efficacy [98]. Although recommended application rates typically avoid acute toxicity, the cumulative impacts of chronic exposure and high-dosage scenarios demand comprehensive investigation [21,37,99]. To mitigate these risks, research into biodegradable, plant-derived carriers including chitosan and lignin-based systems is advancing. Controlled-release formulations, such as hydrogels or polymeric matrices, not only enhance persistence and prophylactic efficacy but also minimize off-target dispersion [35].

While laboratory synthesis of nanosuspensions is well established, scaling these protocols for agricultural deployment demands intricate, resource-intensive processes, including high-pressure homogenization and wet milling [95]. The necessity for precise particle size control, combined with the substantial cost of excipients and solvents, can render such applications prohibitive to smaller agricultural operations [97]. Optimizing nanosuspension formulations necessitates stringent management of particle size distribution, zeta potential, and crystallinity as these parameters susceptible to fluctuation during scale-up. Advanced preservation techniques, notably lyophilization and spray drying, are frequently required to maintain particle stability, increasing both operational complexity and cost [97]. The efficient removal of residual solvents and surfactants remains an unresolved technical bottleneck.

Regulatory oversight of nanopesticides remains incipient, with limited harmonization in protocols evaluating safety, efficacy, and environmental ramifications. This regulatory ambiguity impedes timely product approval and market integration, stalling widescale adoption [95]. Moreover, public apprehension rooted in perceived health risks and opaque technological processes compounds regulatory inertia [25].

Attaining high-fidelity targeting powdery mildew pathogens, while preserving host plant integrity and safeguarding beneficial microbiota, poses a significant technical challenge. Controlled release modalities remain suboptimal, often yielding inconsistent dosing and potentiating pathogen resistance [25]. Although surface engineering strategies, such as attaching polyethylene glycol and ligand conjugation, show promise for enhanced specificity, their translation to agronomic settings is still under development [100].

Nanoparticles interact with plant metabolic networks in multifaceted and sometimes unpredictable manners. While certain formulations can induce systemic resistance, others may disrupt essential physiological processes including nutrient assimilation, photosynthesis, and hormonal signaling. The chronic effects of repeated nanoparticle exposure are insufficiently elucidated and necessitate comprehensive field validation and toxicological assessment [25]. Such interactions are highly contingent upon the physicochemical attributes of the nanoparticles, necessitating tailored, crop-specific investigations.

## 6. Future Perspectives and Directions

Looking ahead, the integration of nanotechnology and precision agriculture is poised to further transform powdery mildew management across diverse cropping systems. The development of smart nanoparticles capable of responding to environmental cues, such as pH, temperature, and pathogen presence, offers the potential for increasingly targeted and timely fungicide delivery. Continued advancements in biodegradable carriers derived from natural polymers like chitosan, cellulose, and starch may enhance plant protection while simultaneously supporting nutrition and growth. The ongoing evolution of digital agriculture, including the adoption of drone-based multispectral sensing and AI-driven diagnostics, is expected to optimize nano-formulation deployment and monitoring, enabling highly site-specific treatments and improved efficiency in antifungal agent application [101,102,103]. Future systems will leverage real-time analytics to support adaptive and sustainable disease management.

The successful translation of laboratory innovations into widespread field applications will require robust, interdisciplinary collaboration among plant pathologists, chemists, toxicologists, and regulatory experts. Future progress hinges on rigorous, large-scale field trials and the harmonization of international regulatory frameworks to confirm safety, efficacy, and environmental impact [59]. Continued research into biodegradable stabilizers, delivery platform optimization, and comprehensive risk assessments remains vital for broader adoption and acceptance. As powdery mildew persists and is exacerbated by changing climate conditions, nano-enabled strategies—such as multi-site fungicides and genomic-assisted breeding will be essential for breeding more resilient host plants. The next wave of advanced tools, including RNA interference and CRISPR gene editing, is anticipated to enable even more precise manipulation of host-pathogen interactions, while real-time diagnostics will inform fungicide use and resistance management.

Emerging smart nano-delivery systems are expected to release antifungal agents in response to specific signals, achieving unprecedented levels of precision. For example, future pH-responsive nanoparticles may more effectively target acidic infection sites, temperature-sensitive carriers could activate exclusively under disease-promoting conditions, and photo-triggered systems might allow for controlled release via light exposure [79]. The functionalization of nanoparticles with selective ligands will likely further enhance specificity and minimize non-target effects. Notably, chitosan-based nanoparticles conjugated with lectins have demonstrated superior efficacy against *Blumeria graminis* spores, hinting at future directions for tailored nanomaterials [79].

The ongoing integration of smart nano-formulations with advanced digital platforms, including drone applications and automated sprayers will drive a continued shift toward precise, sustainable, and compliant powdery mildew management. Future mainstreaming of these innovations will rely on multidisciplinary research, thorough field validation, and the establishment of global guidelines to safeguard food system security. Ultimately, advanced nanotechnology and precision agriculture are expected to deliver transformative, environmentally responsible solutions for powdery mildew control, addressing both present challenges and those on the horizon.

## 7. Conclusions

In conclusion, this review highlights the promising role of nanotechnology in managing powdery mildew across vegetable and fruit crops. Metallic and non-metallic nanoparticles including AgNPs, CuNPs, ZnONPs, SiNPs, and SNPs demonstrate potent antifungal activity through mechanisms such as membrane disruption, ROS generation, and gene regulation, while nano-encapsulated fungicides and essential oils enhance targeted delivery and plant defense activation. These nano-enabled strategies offer synergistic benefits when combined with conventional fungicides, reducing chemical inputs and environmental risks. However, challenges remain, including ecotoxicity, formulation stability, scalability, and regulatory gaps. Future directions emphasize smart delivery systems, biodegradable carriers, and integration with digital agriculture platforms to achieve precision, sustainability, and regulatory compliance. Interdisciplinary research and harmonized safety frameworks are essential to realize the full potential of these innovations in crop protection.

## Figures and Tables

**Figure 1 jof-11-00896-f001:**
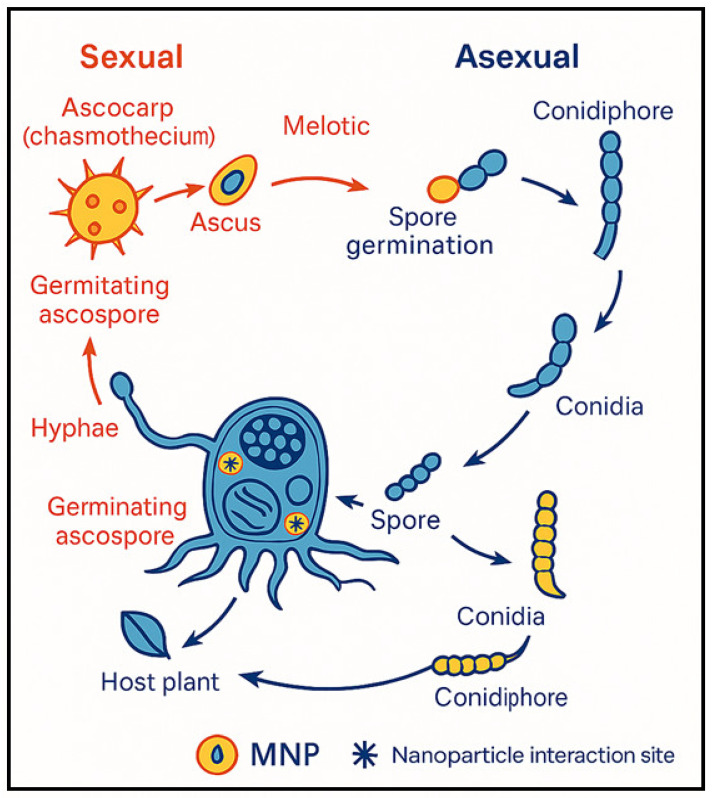
Sexual and asexual phases off powdery mildew life cycle and the interaction site of the nanoparticles within the stage of its life cycle. The figure is generated using Chat GPT 5.

**Table 1 jof-11-00896-t001:** Metallic nanoparticles size, concentration and effectiveness on powdery mildew treatment. Where GS: green synthesis, CS = chemical synthesis.

Nanoparticle Type	Crop	Size (nm)	Concentration	Synthesis Method	Application Method	Effectiveness/Efficiency	Additional Benefits	Reference
Ag	Eggplant	7–25	10–100 ppm	CS	Foliar	Effective in reducing powdery mildew	Mycelial and spore deformation	[35]
Ag	Beans	~25	10–100 ppm	CS and GS	Foliar	Effective against powdery mildew and *Botrytis cinerea*	↓ Disease incidence, ↑ yield potential	[35]
Ag	Melons	7–25	10–100 ppm	CS	Foliar	100 ppm: 20% disease incidence	Spore deformation, ↑ efficacy pre-infection	[35]
Ag	Radish	7–25	10–100 ppm	CS	Foliar	Effective in reducing powdery mildew (extrapolated)	Spore deformation, safe for leafy crops	[35]
Ag	cucumber & pumpkin	~10–50	100 ppm	GS	Foliar	Highest inhibition rate	Damages fungal hyphae and conidia	[35]
Ag	Grapevine	~20–23	crude	GS	Foliar	Improved control of *E. necator*	Superior leaf adhesion, enhanced uptake, prolonged protection vs. copper formulations	[39]
Ag	Grapevine	~17	crude	GS	Foliar	Protective effect against *Uncinula necator*	enhanced sugar, starch, water content; increased shoot length and grape yield	[44]
Cu	Squash	~40–60	~300 mg·L^−1^	CS	Foliar	Highest among tested	Outperformed biological and botanical alternatives; consistent disease suppression	[47]
ZnO	Tomato, Pepper	~40	50–250 mg·L^−1^	CS	Foliar & Soil	Significant reduction in powdery mildew	Enhanced chlorophyll, lycopene, β-carotene, sugar content; reduced oxidative stress	[48]
ZnO	Pepper	79.5	100, 150, 200 mg·L^−1^	CS	Foliar	Significant reduction in disease severity	Increased chlorophyll; no substantial cytotoxicity (mitotic index unaffected); alternative to penconazole	[49]
MgO	Pepper	53	100, 150, 200 mg·L^−1^	CS	Foliar	Significant reduction in disease severity	Increased chlorophyll; no substantial cytotoxicity (mitotic index unaffected); alternative to penconazole	[49]
Fe_3_O_4_	Lettuce	~20–50	~200 mg·L^−1^	CS	Foliar	Significant reduction in disease severity	Increased chlorophyll, carotene, phenolics, protein; elevated antioxidant enzymes activity	[50]
Se	Melons	~50–100	25–75 mg·L^−1^	CS	Foliar	~21–45% reduction	Enhances antioxidant enzymes; alters polyamine, phenylpropanoid, and hormone pathways	[51]
Se	Various crops	~50–100	25–100 mg·L^−1^	GS	Foliar	High antifungal activity; effective against resistant strains	Antioxidant, biocompatible, low toxicity; safe fungicide alternative	[52]
Ag, ZnO, TiO_2_	Tomato	10–100	Varies	GS	Foliar	Effective against fungal & insect pests	↑ Photosynthesis	[53]
CuO and ZnO	Mustard	~50–80	100–300 mg·L^−1^	GS	Foliar	Promising antifungal activity	Eco-friendly alternative to fungicides	[54]
TiO_2_	Spinach	~20	50–100 mg·L^−1^	Sol-gel/GS	Foliar	↑ Photosynthesis, ↓ fungal stress	↑ Biomass, ↑ Chlorophyll content	[54]
CuO	Lettuce	230–400	100 mg·L^−1^	GS	Foliar	↓ Fungal colonization	↑ Leaf health, ↓ oil evaporation	[54]
Ag and ZnO	Grapes	10–50	50–200 ppm	GS	Foliar	Effective against Erysiphe necator	↑ Fruit quality, ↓ chemical residues	[54]
ZnO and CuO	Oranges	20–80	100–300 ppm	GS	Foliar	Antifungal & antibacterial	↑ Shelf life, ↑ Disease resistance	[54]
ZnO	Beetroot (Sugar beet)	—	10, 50, 100 ppm	Engineered	Foliar	↓ Disease severity, ↑ chlorophyll,	Induced resistance via ROS and phenolics	[55]
Ag	Cucurbits	~10–50	25–100 mg·L^−1^	CS/GS	Foliar	Up to 90% reduction	Minimal phytotoxicity; eco-friendly; strong antifungal activity	[8]

Upward arrow = increase, and downward arrow = decrease.

## Data Availability

No new data were created or analyzed in this study. Data sharing is not applicable to this article.

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
