# Peer review of "Harnessing Nanoparticles and Nanosuspensions to Combat Powdery Mildew: A Frontier in Vegetable and Fruit Protection"

_jof, 2025, doi:10.3390/jof11120896_

Round 1
Reviewer 1 Report
This review explores the emerging role of nanotechnology, specifically nanoparticles and nanosuspensions, in managing powdery mildew. This review is of great significance for the prevention and control of plant diseases, especially Powdery mildew. The biggest problem of this paper is that although the title mentions Vegetable and Fruit Protection, wheat is one of the most important crops in the world, and powdery mildew is an important disease of wheat. Why is the prevention and control of wheat powdery mildew not mentioned in the paper?
Dear authors:
In addition, specific details are as follows:
- Line 87, R genes needs italics.
- Line 93, In Resistance genes, "Resistance" should be abbreviated as "R" and in italics.
- Line 87, Integrated Pest Management (IPM) needs to be abbreviated, as mentioned earlier.
- In discussion. The discussion section could benefit from more explicit comparisons to similar international studies, highlighting the novelty and broader applicability of the findings.
- The article has many problems, especially the issues of abbreviations and short forms are particularly serious. If it is not carefully revised, the article will not be accepted.
Author Response
Dear Reviewer,
Thank you very much for taking the time to review our manuscript and give us constructive and insightful comments, which have greatly improved the clarity, depth, and scientific focus of our review. Attached is a detailed, point-by-point response to each question and suggestion provided.
Regards,
Laura Carson

Reviewer 2 Report
There are numerous reports on fungicidal activity of nanoparticles and their potential application in crop protection. The ecotoxicity issues of currently used pesticides make this topic very relevant. The considered review describes current development status of nanoparticle application for vegetable and fruit protection against powdery mildew, one of the most important phytopathogens. This manuscript can be interesting for a wide readership, however, I think it needs substantial improvements prior to acceptance.
- Term “Nano-fungicides” in the present review is associated with nanosuspension or nanoparticles, however, it currently may have a broader meaning (see, for example https://doi.org/10.1016/B978-0-323-95305-4.00001-7, https://doi.org/10.3390/biophysica5020015). Moreover, nanoemulsions, which can be considered as nanopesticides either, represent one of innovative formulations of traditional chemiscal pesticides.
- “The disease powdery mildew thrives in warm, dry climates with high humidity and 42 is especially problematic in greenhouse and field-grown leafy greens.” – “dry climates” contradicts “high humidity”, please, rephrase.
- The statement “Unlike traditional pesticides, the minuscule size of nanoparticles typically ranging from 1 to 100 nm grants them increased surface reactivity. This unique property allows for more efficient delivery and targeted action against pathogens, which in turn means lower doses of active agents are needed.” is questionable, because it is unclear – compared to what? If one considers organic small molecules used widely in crop protection, many of them are highly specific (single site mode of action, as noted in the review) and demonstrate systemic action in contrast to contact action of insoluble particles. Some modes of action typical for nanoparticles, such as s ROS generation, are non-specific. The section “5. Challenges and Limitations” agrees with the correctness of the latter statement. Authors should try to highlight pros and cons of nanoparticles more carfully.
- The nature of nanoparticles (including crystal structure and surface properties, oxidation state of metal) and its impact on activity is almost not considered in the review.
- In many places of the review, it is unclear wether discussed nanofungicides are used in crop production or just were demonstrated to be effective for potential usage, for example: “For example, Vizitiu et al. [40] showed that AgNPs synthesized from Dryopteris filix-mas extract, as well as the extract itself, serve as promising alternatives for controlling Uncinula necator in vineyards.”
- Terms “GS: green synthesis” and “CS=chemical synthesis” seem not precisely charactrrized and thus not scientific enough.
- Provided Tables are very interesting, however, they do not allow to compare the efficiency of reported nanofungicides with some chemical controls.
- In my opinion section “6. Future Perspectives and Directions” should be significantly shortened to be useful. The brevity of the wording could be significantly improved.
Author Response
Dear Reviewer,
Thank you very much for taking the time to review our manuscript and give us constructive and insightful comments, which have greatly improved the clarity, depth, and scientific focus of our review. Attached please find a detailed, point-by-point response to each question and suggestion provided.
Regards,
Laura Carson

Reviewer 3 Report
The manuscript is a nice review of an interesting topic.
On page 15 a section is written in italic.
Author Response
Dear Reviewer,
Thank you very much for taking the time to review our manuscript and give us constructive and insightful comments, which have greatly improved the clarity, depth, and scientific focus of our review. Attached please find point-by-point response to each question and suggestion provided.
Regards,
Laura Carson

Round 2
Reviewer 1 Report
There are no further questions.
There are no further questions.
Author Response
Dear Reviewer,
Thank you very much for taking the time to review our manuscript and give us constructive and insightful comments, which have greatly improved the clarity, depth, and scientific focus of our review. Attached please find the response to the suggestion provided.
Regards,
Laura Carson

Reviewer 2 Report
The manuscript was improved and can be suitable for publication. However, I have to note that it is almost not illustrated and lacks quantitative scientific data. The text sounds a little bit populistic and lacking strong scientific background to a sceptic potential reader.
Section “6. Future Perspectives and Directions” written partially in present simple and present perfect tense contrasts with the review content and answers of Authors to the question 5. This seems to greatly overstate the current state of affairs in the research field.
I still think tables can become much more useful if quantitative efficiency metrics will be added (Question 7).
Target delivery is very interesting. However, authors do not go into details of mobility and permeability of nanoparticles.
Author Response
Dear Reviewer,
Thank you very much for taking the time to review our manuscript and give us constructive and insightful comments, which have greatly improved the clarity, depth, and scientific focus of our review. Attached please find point-by-point response to each question and suggestion provided and corrected manuscript with highlight where changes made.
Regards,
Laura Carson
